

# 2B or not 2B, a study of bottom-quark-philic semi-visible jets

Deepak Kar[1*], Wandile Nzuza[2†] and Sukanya Sinha[3‡]

1 School of Physics, University of Witwatersrand, Johannesburg, South Africa, and
Royal Society Wolfson Visiting Fellow at the University of Glasgow, United Kingdom
2 School of Physics, University of Witwatersrand, Johannesburg, South Africa
3 School of Physics and Astronomy, University of Manchester, United Kingdom

★ deepak.kar@cern.ch , † wandile.nzuza@cern.ch , ‡ sukanya.sinha@cern.ch

## Abstract

Semi-visible jets arise in strongly interacting dark sectors, resulting in jets overlapping with the missing transverse momentum direction. The implementation of semi-visible jets is done using the Pythia Hidden Valley module to mimic the QCD sector showering in so-called dark shower. In this work, only heavy flavour Standard Model quarks are considered in dark shower, resulting in a much less ambiguous collider signature of semi-visible jets compared to the democratic production of all five quark flavours in dark shower. The constraints from available searches on this signature are presented, and it is shown the signal reconstruction can be improved by using variable-radius jets. Finally a search strategy is suggested.

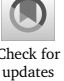

# 1 Introduction

Collider searches for Dark Matter (DM) have so far mostly focused on scenarios where DM particles are produced in association with Standard Model (SM) particles, typically termed *mono-X* searches, where *X* is the SM particle. However, no evidence of DM has been observed so far. Several recent models [1–3] have been proposed that include a strongly-coupled dark sector, giving rise to uncovered collider topologies. Semi-visible jets (SVJ) [4–6] is one such example. In the *t*-channel production mode, a scalar bi-fundamental mediator ($\Phi$) acts as a portal between the SM and the dark sectors, resulting in producing jets interspersed with dark hadrons. At leading order the two SVJs are back-to-back and the direction of the missing transverse momentum is aligned with one of the two reconstructed jets.

Experimental results in the *s*-channel production mode has been presented by the CMS collaboration [7] and in the *t*-channel production mode by ATLAS collaboration [8]. Recent works proposed ways to ascertain systematic uncertainties on SVJ production [9], test the use of jet substructure observables [10, 11], or machine learning methods to understand and discriminate SVJ [12–16]. New signatures of SVJ with leptons [17] or taus [18] have been proposed as well. The recent Snowmass Whitepaper [19] presents a comprehensive review of the underlying theory.

In this paper, we propose a new signature of SVJ only being produced with SM *b*-quarks in the *t*-channel, hereafter referred to as SVJ-b. In Section 2 the details of the model and the signal generation is described. Then Section 3 discusses the possible constraints on this model from existing results. One of the advantages of this signature is the SVJs would be *b*-tagged, so it is easier to identify the SVJ candidate. This allowed us to study several jet reconstruction techniques to compare the efficiency of SVJ reconstruction, which is presented in Section 4. Finally in Section 5, a search strategy is proposed, keeping in mind that certain background processes for a final state with large missing transverse momentum are hard to simulate, especially in particle level. Even though the paper only looks at the *t*-channel production mode, some of the conclusions about jet reconstruction or sensitive observables derived here can also be applicable in a resonance search.

# 2 SVJ with heavy flavour

The strongly coupled dark sector makes use of the so-called *dark shower* (DS), emulating the QCD parton shower. The dark quarks (dark-)hadronise to unstable dark hadrons, which decay partially to SM quarks and partially to stable dark hadrons. The ratio of the rate of stable dark hadrons over the total number of dark hadrons in the event is termed $R_{\text{inv}}$. So far, the experimental analyses have looked in the case where the five SM quarks are produced democratically in the DS. In this paper, we consider the case where the SM component of the dark hadron decay is exclusively to *b*-quarks, as shown in in Figure 1.

The modelling of this final state signature is performed using the Hidden Valley (HV) [20] module of Pythia8 [21], which was designed in order to study a sector which is decoupled from the Standard Model. In HV module, the SM gauge group $G_{sm}$ is extended by a non-Abelian gauge group $G_d$, where the SM particles are neutral under $G_d$, but new HV light particles are charged under $G_d$ and neutral under $G_{sm}$. The interactions between SM fields and the HV particles are allowed by TeV-scale operators. The simplest HV model [22–24] assumes the addition of a $U(1) \times SU(N_d)$ gauge group, with couplings $g'$ and $g_d$, with the $U(1)$ being broken by a scalar $< \phi >$.

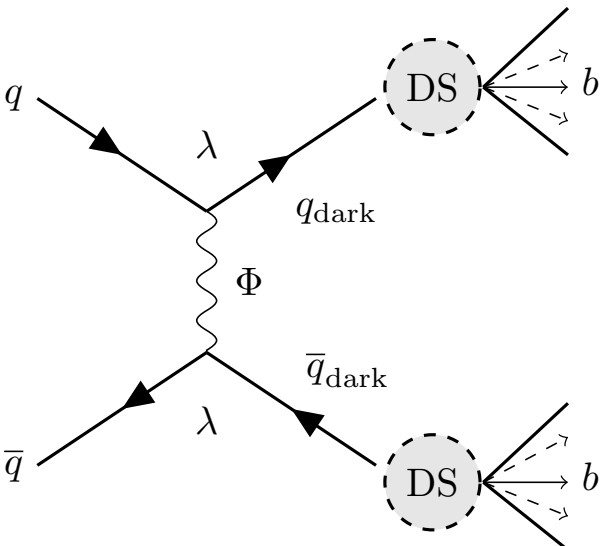

Figure 1: A diagram illustrating the production of semi-visible jets via a $t$-channel mediator, $\Phi$, producing a pair of dark quarks, labelled $q_{\text{dark}}$. DS denotes the dark shower which produces a final state consisting of SM b-hadrons and dark hadrons, governed by the $R_{\text{inv}}$ fraction. The coupling strength of the $q$–$q_{\text{dark}}$–$\Phi$ interaction is denoted by $\lambda$. The figure is adapted from Ref [8].

The number of flavours in the HV module was set to two. This produces vector dark mesons having spin 1 ($\rho_{\text{d}}$) and pseudo-scalar dark mesons having spin 0 ($\pi_{\text{d}}$). The $\rho_{\text{d}}$ decay promptly to $\pi_{\text{d}}$ pairs. The flavour diagonal $\pi_{\text{d}}$ undergoes a decay to SM quarks due to portal interactions of the mediator which couples the SM sector to the dark sector [3]. If the flavour diagonal $\pi_{\text{d}}$ are much lighter than the other HV hadrons, a helicity flipping suppression forces the $\pi_{\text{d}} \to b\bar{b}$ to be dominant, provided the masses satisfy: $2m_{\text{b}} < m_{\pi_{\text{d}}} < 2m_{\text{t}}$, where $m_{\text{b}}$ and $m_{\text{t}}$ denote the bottom and the top quark masses. This is the same helicity flipping suppression as is observed in the case of SM $\pi^+$ decaying to $\mu^+\nu$ instead of $e^+\nu$. The off-diagonal $\pi_{\text{d}}$ remains stable and invisible and contribute to $R_{\text{inv}}$. The finer details of the model parameter choices are currently at an exploratory stage, however this is a simple way to generate this topology, and useful to motivate an experimental search.

The signal samples, at center-of-mass of $\sqrt{s} = 13$ TeV are generated by using a $t$-channel simplified dark-matter model [25] in Madgraph5 [26] matrix element (ME) generator with up to two extra partons at the leading order. The NNPDF3.0LO [27] parton distribution function (PDF) set was used. A mediator mass of 3000 GeV was used, as it has not been excluded by the ATLAS $t$-channel search, but the general conclusions were seen to be valid for higher or lower mediator masses as well. The HV was used to shower the ME-level event and produce dark hadrons. The MLM [28] jet matching scheme, with the matching parameter set to 100 GeV, was employed.

The mass of the dark quark was set to 10 GeV, the flavour-diagonal $\pi_{\text{d}}$ and $\rho_{\text{d}}$ meson masses were set to 20 GeV and 40 GeV respectively, and the off-diagonal $\pi_{\text{d}}$ and $\rho_{\text{d}}$ meson masses were set to 9.99 GeV and 19.99 GeV respectively. These choices are based on References [5,19] and kinematic considerations for enabling the relevant decays as well as to allow for the helicity suppression condition. The general topology of the signal events shows negligible sensitivity to the chosen mass values.

The branching fraction of unstable dark mesons decaying to stable dark mesons, $R_{\text{inv}}$ is a free parameter of the model. Different $R_{\text{inv}}$ fractions result in relatively different kinematics, so representative $R_{\text{inv}}$ values of 0.33 and 0.67 are studied. For HV scenarios, it is convenient to set the overall mass scale of dark hadrons using a non-perturbative definition of QCD confinement scale, and a rough estimate of dark confinement scale $\lambda_{\text{d}}$ can be obtained from lattice calculations as discussed in Ref. [19]. The $\lambda_{\text{d}}$ value was set to 6.5 GeV. The Pythia8 HV $\alpha_{\text{dark}}$ coupling was chosen to be running at one-loop. Another free parameter in the model is the strength of the coupling connecting the SM and DM sectors. The multijet and $t\bar{t}$ processes are generated with Pythia8. As we are mostly concerned with the overall features, the lack of higher order matrix element in background simulation is not a concern. For these studies, the Rivet [29] analysis tool-kit was used, with the Fastjet package [30] for jet clustering.

## 3   Current constraints on SVJ-b signature

It is instructive to consider what constraints the current LHC results with similar final states put on this specific signal. The benchmark signal with mediator mass of 3000 GeV and the $R_{\text{inv}}$ values of 0.33 and 0.67 were considered. Three recent ATLAS results were considered:

1. Search for non-resonant production of semi-visible jets [8].

2. Search for dark matter produced in association with a Standard Model Higgs boson decaying into $b$-quarks [31].

3. Search for supersymmetry in final states with missing transverse momentum and three or more $b$-jets [32].

All the analyses use the full Run 2 ATLAS dataset of 139 fb$^{-1}$ collected at the center-of-mass energy of 13 TeV and they have made their data available in HEPData [33], which allowed to validate the Rivet routines written for the analyses, which are either public or will be soon. The analyses selections were implemented in the particle level, with available smearing for jets applied [34].

The first analysis essentially probes the same topology, but with jets of all flavours. The signal region is defined by no charged leptons, at least two jets, with the leading jet having a $p_{\text{T}}$ of at least 250 GeV. The $H_{\text{T}}$ and $E_{\text{T}}^{\text{miss}}$ was required to be greater than 600 GeV, and at least one jet was required to be within $\Delta\Phi$ of 2 of the $p_{\text{T}}^{\text{miss}}$ direction. The SVJ-b signals were passed through the publicly available Rivet analysis, and yields in the nine-bin signal region (SR) were seen to be negligible compared to data yields, as shown in the Figure 2. This is expected as the $b$-jet production cross-section is small compared to light flavour jets, as well as because the analysis requires less than two $b$-jets in the SR.

For the second analysis, the focus was on the resolved signal regions, as that closely mimics the expected SVJ-b signal topology. The SR definition is rather involved, but essentially probes $E_{\text{T}}^{\text{miss}}$ between 150 to 500 GeV, at requires least no charged leptons, two $b$-tagged jets, none of the jets to be within $\Delta\Phi$ of 20° from $p_{\text{T}}^{\text{miss}}$ direction. The combined $p_{\text{T}}$ of this two-jet system is required to be greater than 100 GeV. The SR is divided into 2 and 2 $b$-tagged regions, and each regions are split into three /metval ranges. The signal yields for a benchmark point with $(m_a, m_A) = 300, 150$ GeV as presented in the auxiliary material in the paper were used to validate our Rivet analysis. In Table 1, the yields for this benchmark signal from our Rivet analysis was compared to yields reported by ATLAS for 2 $b$-tagged and 3 $b$-tagged SRs. Even though the Rivet analysis yields are without acceptance times efficiency corrections, and at particle level, they were mostly within 100%. It is remarkably difficult to reproduce $E_{\text{T}}^{\text{miss}}$

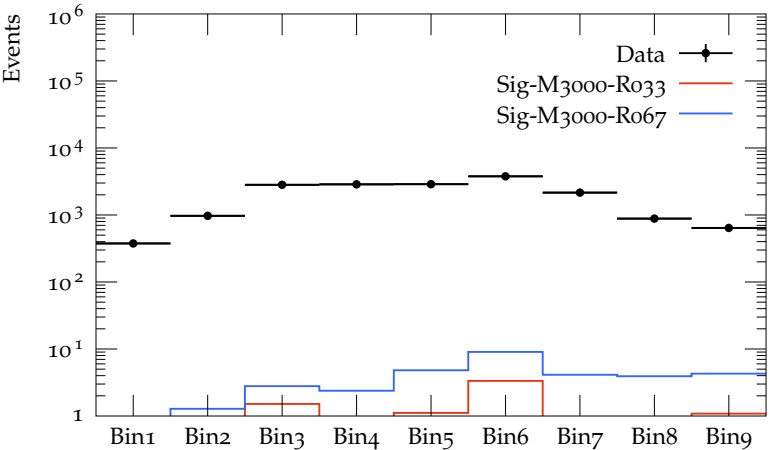

Figure 2: The comparison of yields for benchmark SVJ-b signal in nine-bin SR in [8].

Table 1: Yields in 2b and 3b resolved SR for benchmark signal points, data from [31], and from our Rivet analysis and for SVJ-b signals.

| | Benchmark signal | | Data | SVJ-b signal ($R_{\text{inv}}= 0.33$) | SVJ-b signal ($R_{\text{inv}}= 0.67$) |
|---|---|---|---|---|---|
| Selection | ATLAS yield | our yield | yield | yield | yield |
| **2 *b*-tagged SR** | | | | | |
| $150 \leq E_{\text{T}}^{\text{miss}} < 200$ GeV | 60 | 110 | 14259 | 16 | 12 |
| $200 \leq E_{\text{T}}^{\text{miss}} < 350$ GeV | 70 | 100 | 13724 | 40 | 30 |
| $350 \leq E_{\text{T}}^{\text{miss}} < 500$ GeV | 3.6 | 6 | 799 | 8 | 6 |
| **3 *b*-tagged SR** | | | | | |
| $150 \leq E_{\text{T}}^{\text{miss}} < 200$ GeV | 5.3 | 9 | 408 | 1.1 | 0.8 |
| $200 \leq E_{\text{T}}^{\text{miss}} < 350$ GeV | 18 | 7 | 658 | 6.8 | 2.6 |
| $350 \leq E_{\text{T}}^{\text{miss}} < 500$ GeV | 2.9 | 0.5 | 42 | 0.9 | 0.7 |

distributions at particle level, so this can be considered a sane starting point. The same tables also show data yields and the yields from SVJ-b signals, the latter being much smaller or comparable to both the data and the benchmark signal, which is not excluded by the search. So it is safe to say the considered benchmark point for the SVJ-b signal is not excluded based on this search.

For the third analysis, again a benchmark signal needed to be chosen and the *Gbb* simplified model was picked, as it has the closest final state. The three SRs of interest are defined by requiring no leptons, at least four jets, of which at least three must be *b*-tagged, a minimum $E_{\text{T}}^{\text{miss}}$ of 550 GeV, an effective mass of 1600 to 2600 GeV (defined as the sum of $H_{\text{T}}$ and $E_{\text{T}}^{\text{miss}}$) and $m_{\text{T}}$ of less than 150 GeV. All the jets are required to be $\Delta\Phi$ of 0.4 or further from $p_{\text{T}}^{\text{miss}}$ direction. Here the validation of our Rivet analysis was performed both by comparing distributions after zero lepton pre-selection and yields in specific SRs. The shapes of the distributions were seen to be reasonably similar. The same trend is observed in Table 2, where the yields for this benchmark signal from our Rivet analysis was compared to yields reported by ATLAS for three SRs, denoted by SR-B (boosted), SR-M (moderate) and SR-C (compressed). Again, in absence of acceptance times efficiency correction of the signal in our particle level analysis, our yield can be considered close enough. No SVJ-b signal events passed any of the SR selections, indicating that this search has no sensitivity to SVJ-b signal. This is possibly due to the fact that all *b*-tagged jets are required to be away from the $p_{\text{T}}^{\text{miss}}$ direction.

Table 2: Yields three SRs for the benchmark signal point, data from [32], and from our Rivet analysis and for SVJ-b signals. For the SVJ-b signal, both $R_{\text{inv}}$ values yielded zero events.

| Selection | Benchmark signal | | Data | SVJ-b signal |
| | ATLAS yield | our yield | yield | yield |
|---|---|---|---|---|
| SR-B | 10.13 | 15 | 7 | 0 |
| SR-M | 28.30 | 21 | 18 | 0 |
| SR-C | 34.71 | 22 | 32 | 0 |

Another search, namely the search for heavy particles in the $b$-tagged dijet mass distribution with additional b-tagged jets [35] utilising the same dataset could have been relevant, but since it looks for a resonance mass peak, for our $t$-channel SVJ-b signal, it was seen to have no sensitivity. There can be possible CMS searches, but it was easier for authors to reproduce ATLAS analyses due to the familiarity.

## 4 Signal reconstruction

We use the recent ATLAS $t$-channel search [8] preselection as a starting point, which is listed here. In order to operate at the trigger efficiency plateau of the $E_T^{\text{miss}}$ trigger, a minimum $E_T^{\text{miss}}$ requirement of 200 GeV was needed, although the analysis used a requirement of 600 GeV in the signal region definition. Jets are constructed using the anti-$k_t$ algorithm [36] with a radius parameter of $R = 0.4$, using both charged and neutral inputs. The leading jet $p_T$ was 250 GeV, while all the other jets were required to have $p_T$ of at least 30 GeV. Events needed to have at least two jets. The jet closest to the $p_T^{\text{miss}}$ direction in azimuthal angle was termed the SVJ candidate, and it was required be within $\Delta\Phi < 2.0$ of the $p_T^{\text{miss}}$ direction (this angle will be referred to as $\phi_{\text{MET,closest-jet}}$ subsequently). It was seen that the sub-leading jet was the SVJ candidate in most of the events. Events with charged leptons having $p_T$ of 7 GeV or more were vetoed. Events with two or more $b$-tagged jets were discarded as well, to reduce the background from top quark initiated processes.

Owing to the nature of the current signal, certain modifications have to be made. The 200 GeV $E_T^{\text{miss}}$ requirement severely reduces the signal statistics in particle level, so in order to investigate the strategies to increase the signal efficiency, we have removed $E_T^{\text{miss}}$ requirement. Since this requirement would have been enforced before the use of jets, this does not bias the subsequent studies. The actual experimental analysis will naturally have to use a well motivated $E_T^{\text{miss}}$ threshold. At detector level, mis-measurement of jets typically increase the $E_T^{\text{miss}}$, so this is a reasonable assumption for a particle level study. The lepton veto with the lowest possible $p_T$ was appropriate for the analysis dominated by light-quark initiated jets, however semi-leptonic decays from $b$-quarks indicate the events will have leptons, so we have to optimise that selection as well.

In order to arrive at an analysis strategy which can maximise the signal efficiency for this specific final state, it is worthwhile to investigate different jet reconstruction strategies. This final state offers a clean playground, as the SVJs need to $b$-tagged, unlike for the democratic production of all five flavours where the only handle we have is the azimuthal separation from the $p_T^{\text{miss}}$ direction, which can introduce ambiguities.

- The ATLAS analysis uses anti-$k_t$ jets with radius parameter of 0.4. The use of $E_T^{\text{miss}}$ trigger allows the search to use jets of $p_T$ of 30 GeV, which would not be possible with lowest un-prescaled single jet triggers, which currently in ATLAS requires a leading jet $p_T$ of 450 GeV [37].

- As semi-visible jets tend be spread out with *gaps* in them, in our previous work [11], we investigated the use of large-radius jets and jet substructure observables. The jet energy calibration for large-radius jets with anti-$k_t$ algorithm using a radius parameter of 1.0 with trimming [38] as used in ATLAS [39] typically require a minimum $p_T$ of 200 GeV. We would then require two $b$-tagged large-radius jets with $p_T$ of at least 200 GeV. The $b$-tagging of large-radius jets in ATLAS are typically performed with associated track-jets [40]. While we can still use $p_T^{\text{miss}}$ trigger with large-radius jets, the $p_T$ requirement of 200 GeV for these jets would remain.

- Another possible option will be to use reclustered (RC) jets [41], as that allows us to use already calibrated lower $p_T$ jets than usual large-radius jets. However in the cases where the RC jet consists of only one small-radius input jet, the experimental jet mass calibration tends to be ill-defined. We have observed that this is indeed the case for a significant number of events. Since the radius is fixed, it is difficult to avoid this problem.

- Jet reconstruction with a variable radius (VR) [42] was introduced to increase the signal reconstruction efficiency for boosted resonance searches. The VR algorithm needs the minimum and maximum allowed jet radius, as well $\rho$, which is the mass-like parameter, resulting in the effective radius of the VR jets to scale as $(R\rho/p_T)$. While different choices can be made for input jets to VR algorithm, starting from track-jets (jets only with charged particles at particle level) where going to much lower $p_T$ is feasible. However the track-jets do not capture the totality of the SVJ, which we saw leads to non-optimal performance for signal to background discrimination later. We therefore used anti-$k_t$ jets with radius parameter of 0.4 with a minimum $p_T$ of 30 GeV as input to VR algorithm.

  The minimum radius was kept at 0.4 and the maximum was set to 1.2 to stay well within the central part of the detector. The suggested value of $\rho$ is $< 2p_T$ in case of resonance searches, here we used 500 GeV, which is consistent with using a leading jet $p_T$ of 250 GeV. It was checked that using a lower value did not affect the shape of kinematic distributions, but reduced the acceptance. This can be understood from the fact that SVJ indeed behaves like a multi-prong jet [11]. Since VR jet radius is adaptive compared to RC, we decided to use VR jets over RC jets. ATLAS has used VR tracks jets in previous searches [31].

- Jet reconstruction with Dynamic Radius Jet Clustering Algorithm (DR) [43] allows the radius to vary dynamically based on local kinematics and distribution in the angular plane inside each evolving jet. It modifies the fixed radius by an additional term, which captures the $p_T$-weighted standard deviation of the distances between pairs of fundamental constituents of an evolving pseudojet (i.e. the intermediate jet). This can be useful for SVJ signal as wider radiation pattern makes these jets larger, but not uniformly larger, as the use of VR jets showed.

In order to compare the performance of the above mentioned jet reconstruction strategies, we need to come up with a metric. As the pre-selection will require a $b$-tagged SVJ close to $p_T^{\text{miss}}$ direction, after requiring at least two $b$-tagged SVJ, we demand at least one of them must be within $\Delta\Phi < 2.0$ of the $p_T^{\text{miss}}$ direction, without requiring any higher leading jet $p_T$ threshold. Then in Figure 3, we look at the multiplicity of $b$-tagged jets for jets with radii 0.4 and 1.0, as well as of VR and DR jets, before any pre-selection.

It is evident that the use of VR jets will enhance the signal efficiency, as illustrated in the Table 3 as well. DR jets also show a dramatic shift toward higher jet multiplicities, which will be discussed later in this section.

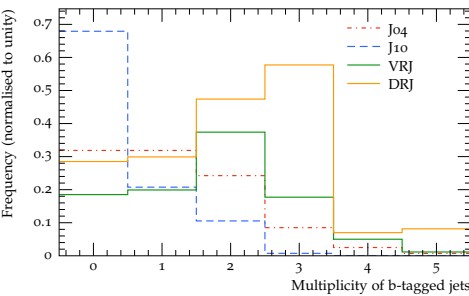 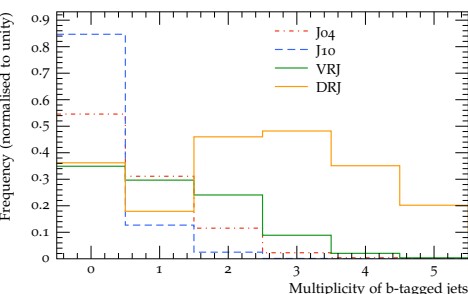

Figure 3: The multiplicity of $b$-tagged jets for radius 0.4 (J04) and radius 1.0 jets (J10), as well as VR jets (VRJ) and DR jets (DRJ), for signals with $R_{\text{inv}}$ of 0.33 (left) and 0.67 (right). The J04, J10, VRJ and DRJ respectively refer to jets with radius 0.4, 1.0 and VR and DR jets.

Table 3: The efficiency of signal selection for using jets with radius 0.4, 1.0 and VR and DR jets.

|  | Selection Efficiency in % | |
| --- | --- | --- |
| Selection | Signal $R_{\text{inv}} = 0.33$ | Signal $R_{\text{inv}} = 0.67$ |
| J04 | 33 | 12 |
| J10 | 11 | 3 |
| VRJ | 60 | 35 |
| DRJ | 81 | 66 |

Requiring further kinematic thresholds, such as requiring a $p_{\text{T}}$ of 250 GeV of the leading jet, or $H_{\text{T}}$ of 600 GeV as in the SR definition, reduces these numbers proportionally. The reason is clear, for anti-$k_t$ jets with radius 1.0, the experimentally required $p_{\text{T}}$ threshold of 200 GeV kills a large fraction of the signal. The smaller efficiency of anti-$k_t$ jets with radius of 0.4 compared to VR jets can be attributed to the fact that for events where DS decay products are more spread out, it does not encompass everything in the fixed smaller radius cone.

In order to understand the above mentioned signal efficiency gain with VR, we look at the objects in $\eta - \phi$ plane for four representative events in Figure 4. VR algorithm is seen to reconstruct the SVJs better than jets with radius of 0.4, both in terms of containing all the decay products, as well as being closer to the direction of missing transverse momentum. This is found to be true for both the signal points with rather different $R_{\text{inv}}$ fractions. The last event merits a special discussion. This is an example event which will fail the requirement of two anti-$k_t$ $b$-tagged jets with radius of 0.4, but will have two $b$-tagged VR jets. Even though the inputs for VR are anti-$k_t$ jets with radius of 0.4, in this case, there were no $b$-tagged jets fulfilling the $p_{\text{T}}$ requirement, but such non-$b$-tagged jets allowed the VR jet to form, which was then classified as $b$-tagged.

In order to validate this approach, resonant SVJ signals were generated using a $Z'$ mediator of the same mass of 3000 GeV but only allowing SM bottom quarks in the HV decay. The same two $R_{\text{inv}}$ values were used, and it can be seen in Figure 5 that using VR jets, as compared to jets with radius of 0.4 and 1.0, retains significantly more signal, as well as leads to reconstruction of the resonant peaks at appropriate mass values, especially for lower values of $R_{\text{inv}}$ where these searches tend to be more sensitive.

It must be noted that while signal reconstruction efficiency was improved with VR jets, for the leading background processes, which are discussed in the next section, the selection efficiencies were rather similar. So indeed use of the VR jets will improve the signal reconstruction efficiency compared to background processes.

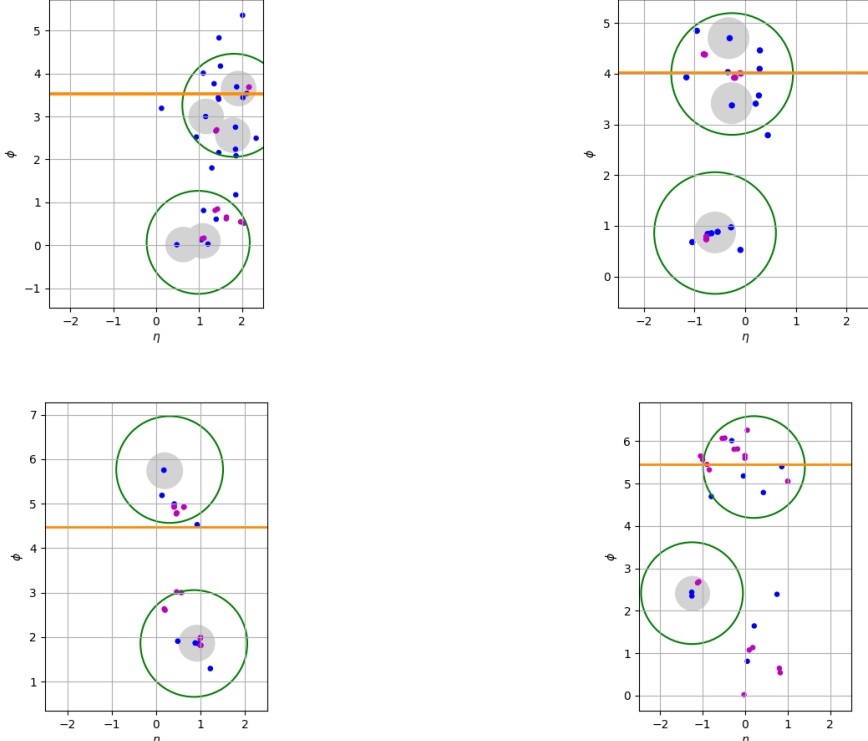

Figure 4: Various objects are plotted in $\eta - \phi$ plane for four representative signal events with $R_{\text{inv}}$ of 0.33 (top row) and 0.67 (bottom row). The large hollow green circles represent the $b$-tagged VR jets, the filled gray circles represent $b$-tagged anti-$k_t$ jets with radius 0.4, the magenta points represent dark hadrons, the blue points indicate stable b-hadrons, and the orange line the direction of missing transverse momentum.

Use of DR jets is also explored, and showed a remarkable increase in efficiency. This can be understood by looking at the $b$-tagged jet multiplicity distrFigure 3. However it must be noted, that unlike VR jets, here the input is the original inputs for the jets. Experimental calibration of arbitrary radius jets are not supported, and in Figure 6, we see the effective radius is about 0.6 to give this high efficiency, which can not be achieved by using radii of 0.4 jets as inputs. Experimentally, only fixed radius jets are calibrated and scale factors are derived to account for difference in data and simulation. Typically the jet constituents individually are not calibrated, and efforts to perform such calibration for jet substructure measurements in ATLAS yielded large systematic uncertainties associated with cluster energy scale and angular resolution, cluster splitting and merging [44, 45]. Therefore we do not think it is experimentally feasible yet to use DR jets.

The lepton veto requirement from the ATLAS analysis was revisited to accommodate the possibility of semi-leptonic decay of $b$-quarks. The pre-selections included discarding events with charged leptons having $p_T$ of 7 GeV or more. While that is reasonable for a mostly light-flavour quark dominated signature, semi-leptonic decay of $b$-quarks produce a copious amount of charged leptons. Even in a particle level study, imposing a lepton veto leads to a loss of about 50% of signal events with the lepton $p_T$ threshold of 7 GeV, and drops to roughly 25% for a higher lepton $p_T$ threshold of 30 GeV, which is non-ideal.

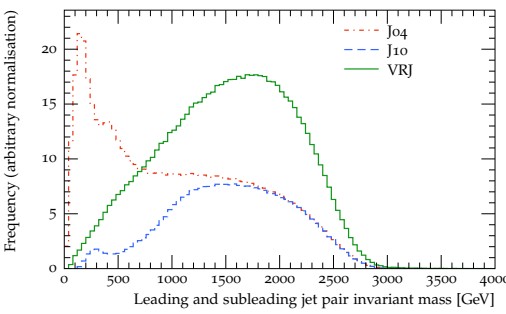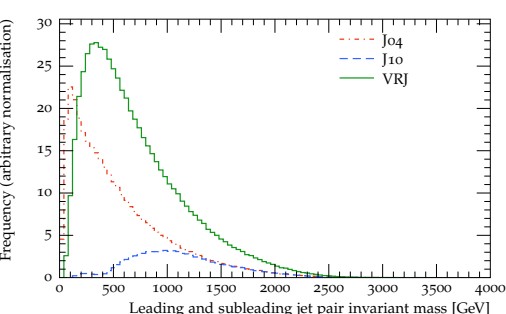

Figure 5: The invariant mass of leading and subleading $b$-tagged jets for radius 0.4 (J04) and radius 1.0 jets (J10), as well as VR jets (VRJ), for $s$-channel SVJ-b signals with $R_{\text{inv}}$ of 0.33 (left) and 0.67 (right).

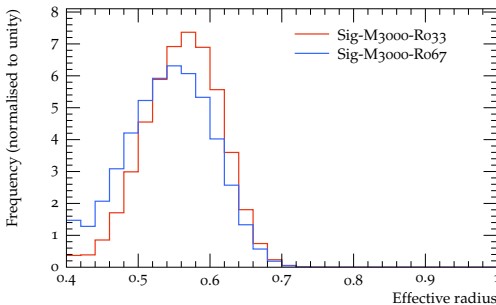

Figure 6: The effective radius given by the DR algorithm for two benchmark SVJ-b signals.

To recover events from lepton veto, two approaches can be adopted, either vetoing events with charged leptons with a significantly higher $p_{\text{T}}$ threshold, or requiring them to be close to the $b$-jet. In Figure 7, the correlation between these two quantities are shown. It can be seen most leptons have low $p_{\text{T}}$ and are close to a $b$-tagged jet. Based on this, we decided to reject leptons only with $p_{\text{T}} > 75$ GeV, and further require they lie within $\Delta\Phi > 0.1$ of a $b$-tagged jet. This leads to no signal efficiency loss, but would still reject a large fraction of background events with an isolated high $p_{\text{T}}$ lepton. While this was found to be the simplest approach in a particle level study, in an experimental analysis, isolation requirements can be employed as well.

# 5 Search strategy

The leading background processes are multijet, $Z$ boson decaying invisibly after being produced with $b\bar{b}$ (referred to as $Z(\nu\nu)b\bar{b}$) and top quark pair production (referred to as $t\bar{t}$). The semi-leptonic and di-leptonic decay modes of $t\bar{t}$ were seen to produce similar kinematic distributions and yields, so we combined them together in non-hadronic mode.

Two approaches were explored. The first was to design an experimental search with specific cuts. However this final state has non-trivial dependence on detector effects, as missing transverse momentum arises both from actual undetected particles as well as from jet energy and angle mis-measurements. The latter is notoriously difficult to model even with full detector simulation [46] and various alternative approaches have been proposed [47]. In this study, as mentioned earlier, the $E_{\text{T}}^{\text{miss}}$ is calculated using smeared jets, but we only treat that as

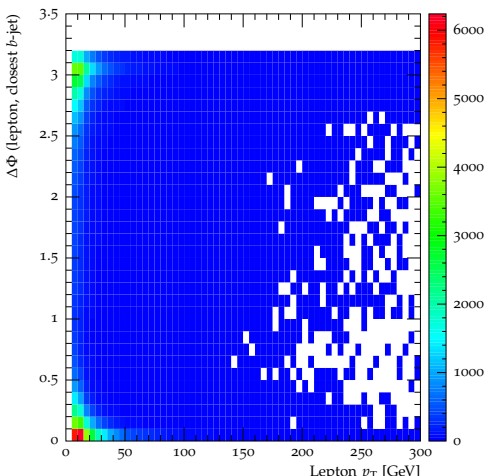

Figure 7: The correlation of charged lepton $p_{\mathrm{T}}$ against the $\Delta\Phi$ distance from the closest $b$-tagged jet is shown for the SVJ-b signal a mediator mass of 3000 GeV and $R_{\mathrm{inv}}$ of 0.33. Same trend is observed for the SVJ-b signal with $R_{\mathrm{inv}}$ of 0.67.

Table 4: A table showing yields for two SVJ-b signals (corresponding to a mediator mass of 3000 GeV and $R_{\mathrm{inv}}$ of 0.33 and 0.67) and all leading background processes for a representative set of cuts. The numbers are normalised to 139 fb$^{-1}$ of integrated luminosity.

| | Yields after each selection step | | | | | |
|---|---|---|---|---|---|---|
| Selection | Signal $R_{\mathrm{inv}}$= 0.33 | Signal $R_{\mathrm{inv}}$= 0.67 | Multijet | $t\bar{t}$ (had) | $t\bar{t}$ (non-had) | $Z(\nu\nu)bb$ |
| Start | 112439 | 112439 | $3 \times 10^{16}$ | $3.9 \times 10^{6}$ | $1.2 \times 10^{5}$ | $1.4 \times 10^{5}$ |
| Lepton Veto | 110521 | 111506 | $3 \times 10^{15}$ | $3.7 \times 10^{6}$ | $1 \times 10^{5}$ | $1.4 \times 10^{5}$ |
| Two $b$-tagged VR jets | 65615 | 38234 | $1.5 \times 10^{14}$ | $3.4 \times 10^{6}$ | $9.4 \times 10^{4}$ | 22640 |
| $\phi_{\mathrm{MET,closest\text{-}jet}} < 2$ | 38824 | 15783 | $6 \times 10^{13}$ | $2.4 \times 10^{6}$ | $4.5 \times 10^{4}$ | 2626 |
| $E_{\mathrm{T}}^{\mathrm{miss}} > 600$ GeV | 1604 | 1316 | 0 | 0 | 292 | 0.1 |
| $H_{\mathrm{T}} > 600$ GeV | 1246 | 673 | 0 | 0 | 208 | 0.1 |

a reasonable approximation. Keeping that in mind, we show that a strict set of cuts, as shown in Table 4 can reduce leading background contributions more drastically than signal. More specifically, this meant applying a $H_{\mathrm{T}}$ and $E_{\mathrm{T}}^{\mathrm{miss}}$ requirement of 600 GeV each. This is somewhat motivated by the ATLAS $t$-channel SVJ search [8], and is needed to be studied carefully in an experimental analysis. A combination of these requirements can also be used to define a signal region (SR). No trigger considerations have been applied here, but a di-$b$-tagged jet trigger or a $E_{\mathrm{T}}^{\mathrm{miss}}$ trigger will probably be the best.

Alternatively we propose some possible observables that can potentially reduce the background contributions. The extremely high yield of multijet before the cuts and vanishing yield after the cuts lead us to focus on the shape differences, so all the distributions are area-normalised.

The discriminating variables that were found to be sensitive are:

- $H_{\mathrm{T}}$, defined as the scalar sum of $p_{\mathrm{T}}$ of the $b$-tagged VR jets in the event.

- $E_{\mathrm{T}}^{\mathrm{miss}}$, Magnitude of missing transverse momentum.

- Effective mass, defined as scaler sum of $E_{\mathrm{T}}^{\mathrm{miss}}$ and $H_{\mathrm{T}}$.

- Lead jet $p_{\mathrm{T}}$, the $p_{\mathrm{T}}$ of the leading $b$-tagged VR jet.

- Number of subjets of radius 0.2 which can formed using $k_t$ algorithm from the leading $b$-tagged VR jet.

- $\phi_{\text{MET,closest-jet}}$, defined as the $\Delta\Phi$ of the $b$-tagged VR jet closest to the direction of the missing transverse momentum.

- $p_{\text{T}}^{\text{bal}}$, $p_{\text{T}}$ balance between the closest jet and farthest $b$-tagged VR jets from $p_{\text{T}}^{\text{miss}}$ in azimuthal direction, termed $j_1\, j_2$, defined as,

$$p_{\text{T}}^{\text{bal}} = \frac{|\vec{p_{\text{T}}}(j_1)+\vec{p_{\text{T}}}(j_2)|}{|\vec{p_{\text{T}}}(j_1)|+|\vec{p_{\text{T}}}(j_2)|}\,.$$

- $\log(M_{\text{bb}}/p_{\text{T}}^{\text{bb}})$, defined by logarithm of invariant mass of the two $b$-tagged VR jets over their invariant $p_{\text{T}}$.

The observables shown in Figure 8 and in Figure 9 can help to reduce the background in the SR, and can be used in a fit to determine the background. The $H_{\text{T}}$ and $E_{\text{T}}^{\text{miss}}$ distributions are rather correlated, but it is seen background processes die off much faster than the SVJ-b signals, which is consistent with cutflow table presented above. The effective mass and the leading jet $p_{\text{T}}$ show a similar trend as well, confirming the fact that the SVJ-b events tend to more energetic. The leading background processes behave similarly for these observables. The number of subjets is an interesting observable. While signal tends to have slightly higher multiplicity compared to multijet background, the $t\bar{t}$ background processes, specially the handronic mode has a high multiplicity as well, which is expected. For $p_{\text{T}}^{\text{bal}}$, while the SVJ-b signal with higher invisible fraction is more balanced than most of the background processes, the signal with lower invisible fraction behaves like hadronic $t\bar{t}$ background process. The latter signal however is the most highly peaked at the lower $\phi_{\text{MET,closest-jet}}$ value. However for $Z(\nu\nu)b\bar{b}$ process $\phi_{\text{MET,closest-jet}}$ is useful discriminating observable, as the $Z$ boson typically balances the $b\bar{b}$ system. The $\log(M_{\text{bb}}/p_{\text{T}}^{\text{bb}})$ distribution does not look great for discriminating $t\bar{t}$ background processes, it can be promising for multijet. However it must be pointed out that none of them individually are sufficient to discriminate between signal and background contributions.

Finally, while individual jet substructure observable can be used in such a fit as shown in [11], or a in ML algorithm, Lund jet plane [48] was shown to be a promising observable. Along the same line, Lund subjet multiplicity [49, 50], which counts the number of subjets above a specified transverse momentum requirement in a jet's angle-ordered clustering history, was looked at in Figure 10. Only multijet background is shown to contrast the signal with light quarks or gluons. The lower values of emission $k_t$ requirements seemed more sensitive to the difference, which is indicative of the fact that SVJs do not contain multiple dominant directions of energy flow, rather *holes* separated by softer emissions.

# 6 Summary

A feasibility study for a collider search for SVJ produced in association with only heavy flavour quarks is presented. While this is a theoretically well motivated scenario, no search has been performed yet. We show that it is a rather promising search channel, and the signal has not been excluded by current searches. The additional requirement of SVJs being $b$-tagged acts as a powerful tool to isolate signal-like events. The use of variable radius reclustered jets and dynamically reclustered jets have been studied, and seen to improve the signal acceptance. Experimentally the former is easier to implement. Finally a search strategy is proposed, as we showed that requiring high $H_{\text{T}}$ and $E_{\text{T}}^{\text{miss}}$ can result in having a good signal over background significance. Several potentially discriminating observables are proposed as well in order to aid the experimental search.

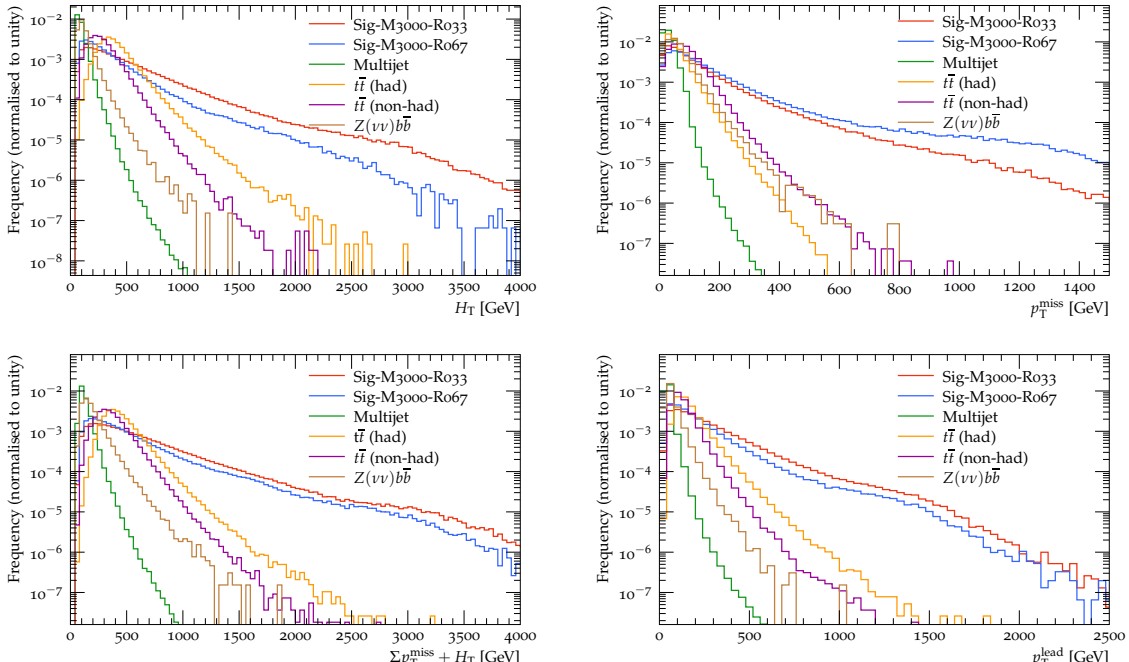

Figure 8: The area-normalised distributions of $H_{\mathrm{T}}$ (top left), $p_{\mathrm{T}}^{\mathrm{miss}}$ (top right) and effective mass (bottom left) and leading jet $p_{\mathrm{T}}$ (bottom right) for two SVJ-b signals (corresponding to a mediator mass of 3000 GeV and $R_{\mathrm{inv}}$ of 0.33 and 0.67) and all leading background processes after only jet multiplicity and lepton angle requirements.

## Acknowledgments

We would like to thank Tim Cohen and Matt Strassler for useful inputs. We are grateful to Tousik Samui for providing us with dynamic radius jet clustering code and to Biswarup Mukhopadhyaya and Ritesh Singh for related discussions. We also thank Cesare Cazzaniga and Suvam Maharana for additional discussions.

**Funding information** DK thanks the generous support by Wolfson Foundation and Royal Society to allow him to spend his sabbatical year at the University of Glasgow. WN is supported by SA-CERN excellence bursary. SS is supported by European Research Council grant REALDARK (grant agreement no. 101002463).

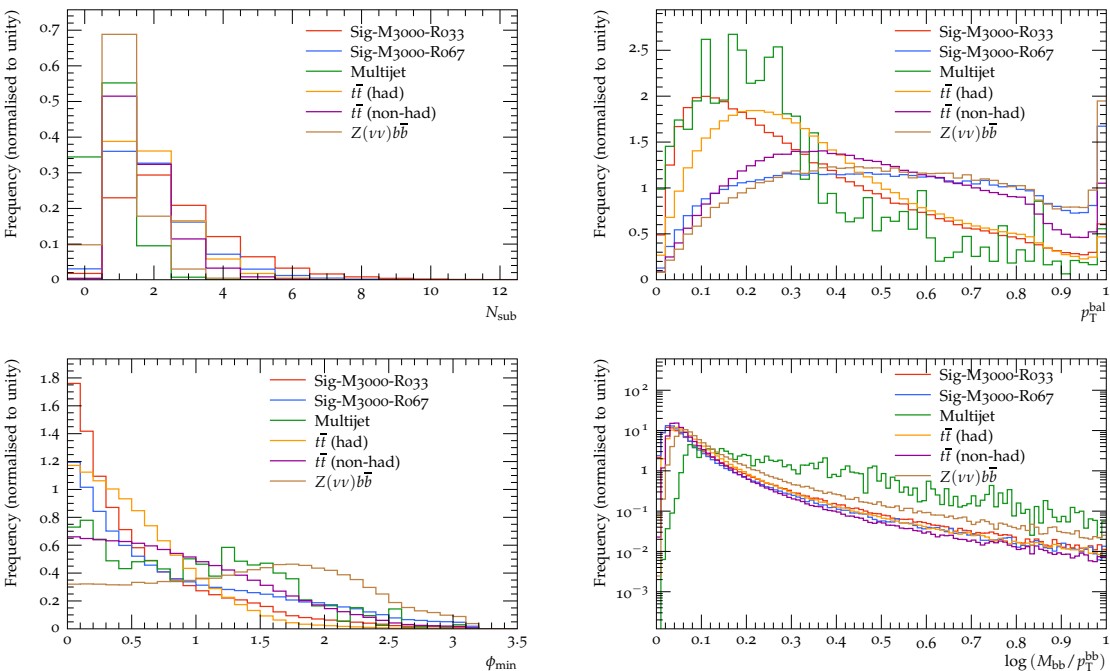

Figure 9: The distributions of number of subjets (top left), $p_T^{\text{bal}}$ (top right), $\phi_{\text{MET,closest-jet}}$ (bottom left) and $\log(M_{\text{bb}}/p_T^{\text{bb}})$ (bottom right) for two SVJ-b signals (corresponding to a mediator mass of 3000 GeV and $R_{\text{inv}}$ of 0.33 and 0.67) and all leading background processes after only jet multiplicity and lepton angle requirements.

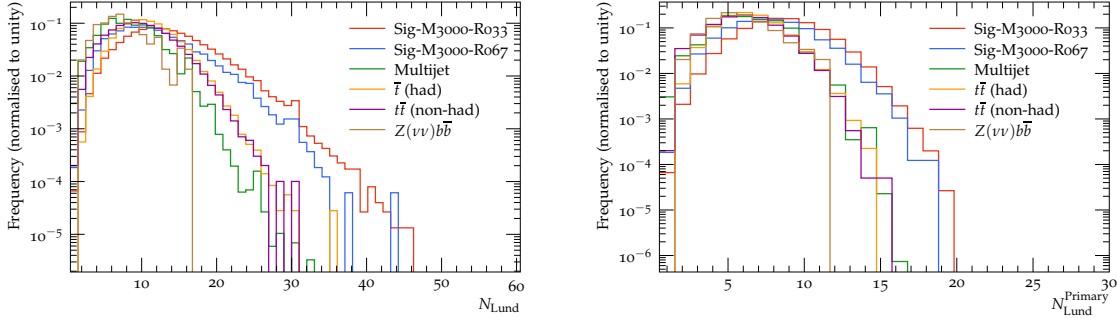

Figure 10: The distributions of number of Lund subjets in full plane (left), and along the primary clustering sequence (right) with an emission $k_t$ requirement of 10 GeV for two SVJ-b signals (corresponding to a mediator mass of 3000 GeV and $R_{\text{inv}}$ of 0.33 and 0.67) and all leading background processes after only jet multiplicity and lepton angle requirements.

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
