# Peer review of "B or not 2B, a study of bottom-quark-philic semi-visible jets"

_SciPost Physics Core, doi:SciPost Phys. Core 7, 071 (2024)_

## Round 2 · Referee Report · Anonymous (Referee 1) · 2024-5-22

Strengths
1- The topic of semi-visible jets is very relevant and of great interest for LHC experimentalists. 2- The specific model, dark pions decaying into bottom quarks, has not been considered previously. 3- The discussion of jet clustering and jet tagging is quite useful.
Weaknesses
1- The paper is not very pedagogical. The introduction is very short and the outline is missing. Subsequent sections are quite technical and written for an expert audience. 2- The model is not explained clearly enough. 3- The purpose of the third section is unclear. 4- The paper ends too early. There should be a recommended set of cuts and an estimate of the resulting signal-to-background ratio.
Report
The manuscript is quite short and feels very rushed. The introduction skips over the general motivation and comes straight to the point, but does not outline the remainder of the work. The second section briefly introduces the model, skipping many details and assuming that the reader is very familiar with the literature. The third section shows that it is difficult to reproduce signal yields for existing analyses, but concludes that these details don't matter because the existing analyses are expected to be blind to this model.
The main point is finally to propose a new analysis strategy that would be better suited for constraining the model under consideration. The main focus is on identifying promising jet tagging algorithms and kinematic variables that may be used for background suppression. However, no actual conclusion is reached regarding the signal-to-background ratio and whether such an analysis could actually be sensitive to the model.
Unfortunately, the paper in its current form does not meet the requirements for publication. A publication could be considered, if a substantial amount of detail is added to the various sections, as listed below.
Requested changes
1- An outline needs to be added to the introduction. 2- In the second section, the structure of the confined phase does not become clear. If there is only one flavour, how can there be several different dark pions? It does also not become clear which dark mesons are assumed to be stable and which ones are assumed to decay. What is the point of the very peculiar choices of the masses of the off-diagonal dark mesons? What is the reasoning behind the choice of $R_\text{inv}$? I understand that the authors are not interested in the model-building details for the purpose of this work, but they need to at least motivate the choices that they make. 3- I don't understand the statement "The dark QCD confinement scale value was set to 6.5 TeV". Isn't that in contradiction with the masses of the dark mesons? Presumably the authors want to say something different here. 4- The proposed model comes with several free parameters, such as the mass of $\Phi$ and its coupling. Without specifying these parameters, the signal predictions stated in the following sections are meaningless. Whether the model is allowed or excluded must surely depend on the mass and coupling of the mediator? 5- In section 3 the authors need to explain more clearly where the differences between the official yields and the simulated yields for the benchmark models stem from and which steps have been taken to improve agreement. 6- For the second analysis (MET+Higgs), it would be interesting to see a histogram of $m_{bb}$ compared to data. Ideally, this should be done for different dark meson masses, to see if one might see a peak in the invariant mass distribution. 7- In section 5 the authors need to propose a combination of cuts with which they can conceivably achieve interesting signal-to-background ratios. At the very least they need to demonstrate that the resulting sensitivity should be much better than for the analyses considered in section 3. 8- The summary needs to contain a clearer conclusion and plan for future work. What are the next steps towards applying such an analysis to real data? 8- In figure 1, one of the circles is labeled "DS", the other is labeled "HV". Why? 9- The purpose of the right panel of figure 2 is unclear. Do the authors have permission to use this figure?
Recommendation
Ask for major revision
We thank the referee for a comprehensive review, and making suggestions to improve the quality and usefulness of the paper. We apologise for the delay in getting back, summer travels came in the way. We tried to address most of the concerns, please find the details inline. This included modifying the model part to make it cleared and adding details of the analyses used in reinterpretation. Additionally, we have added a study of a new jet reconstruction technique called the Dynamic Radius Jet Clustering, added plots of a couple of new discriminating variables, and proposed a loose set up cuts for an experiment analysis with a cutflow.
1- The paper is not very pedagogical. The introduction is very short and the outline is missing. Subsequent sections are quite technical and written for an expert audience.
The introduction now contains an outline to guide the reader, and in general we carefully over the text and tried to make the flow better.
2- The model is not explained clearly enough.
We now started the signal generation discussion with the vanilla SVJ first, before moving to our specific signature and generally tried to make this part clearer.
3- The purpose of the third section is unclear.
We wanted to explain that as far as current experimental results go at the time of preparing the paper, our signal is not excluded. If you prefer us to phrase it differently, we are happy to.
4- The paper ends too early. There should be a recommended set of cuts and an estimate of the resulting signal-to-background ratio.
We expanded the search strategy section, please see below.
1- An outline needs to be added to the introduction.
Please see above, done now.
2- In the second section, the structure of the confined phase does not become clear. If there is only one flavour, how can there be several different dark pions? It does also not become clear which dark mesons are assumed to be stable and which ones are assumed to decay. What is the point of the very peculiar choices of the masses of the off-diagonal dark mesons? What is the reasoning behind the choice of Rinv? I understand that the authors are not interested in the model-building details for the purpose of this work, but they need to at least motivate the choices that they make.
We had accidentally generated it with N_F=1, but now have changed to N_F=2 and the kinematic distributions remain unaffected. There are minuscule differences in the yields for the different recasts. We rewrote this section to explain the HV decay mode clearly. There are unstable (flavour diagonal) and stable (off-diagonal) dark pions, the latter being the DM candidate, and contributing to missing transverse momentum. The mass choices are basically motivated to kinematically allow the decays, again we added that in the text.
3- I don't understand the statement "The dark QCD confinement scale value was set to 6.5 TeV". Isn't that in contradiction with the masses of the dark mesons? Presumably the authors want to say something different here.
This is based from snowmass study: and there was a typo, its 6.5 GeV, mea culpa.
4- The proposed model comes with several free parameters, such as the mass of Φ and its coupling. Without specifying these parameters, the signal predictions stated in the following sections are meaningless. Whether the model is allowed or excluded must surely depend on the mass and coupling of the mediator?
We did mention it in the previous section, but totally understand that it got lost in the details. Added at the beginning of the section, and expanded the tables and the plot with both Rinv values.
5- In section 3 the authors need to explain more clearly where the differences between the official yields and the simulated yields for the benchmark models stem from and which steps have been taken to improve agreement.
In general we agree that neither are great, which is a consequence of modeling large MET as we mentioned. For these kinds of signals, large MET comes from two sources: the real MET from the DM candidates, and the angular and energy mismeasurement of jets. While the first can be modelled well in a pure particle level study, the latter cannot be. One can use Delphes to mimic some sort of detector effect, but in our experience it is far from accurate. We used jet energy and angular smearing implemented in Rivet as we mentioned, which somewhat helps, but we think it is fair to point out the shortcoming.
6- For the second analysis (MET+Higgs), it would be interesting to see a histogram of ššš compared to data. Ideally, this should be done for different dark meson masses, to see if one might see a peak in the invariant mass distribution.
We tried two different masses, the defaults and the twice. The stats get bad with higher MET ranges, but these are the three 2-bagged MET ranged Mbb plot, we cant see any obvious structure. Please see the attached plots.
7- In section 5 the authors need to propose a combination of cuts with which they can conceivably achieve interesting signal-to-background ratios. At the very least they need to demonstrate that the resulting sensitivity should be much better than for the analyses considered in section 3.
The point is well taken, and we added a basic cutflow, with the disclaimer that it will have to be optimised properly in an experimental analysis.
8- The summary needs to contain a clearer conclusion and plan for future work. What are the next steps towards applying such an analysis to real data?
We expanded the summary as suggested.
8- In figure 1, one of the circles is labeled "DS", the other is labeled "HV". Why?
Sorry for that, fixed now.
9- The purpose of the right panel of figure 2 is unclear. Do the authors have permission to use this figure?
The Fig 2 (right) shows the benchmark point considered is not excluded by the ATLAS analysis. Any ATLAS published figure is freely usable (Reproduction of the article, figures and tables are allowed as specified in the CC-BY-4.0 license). However, we realise it is not strictly useful to the flow of the paper, so has been removed.

Author: Deepak Kar on 2024-09-12 [id 4774]
(in reply to Report 2 on 2024-06-10)>>We thank the referee for a comprehensive review, and making suggestions to improve the quality and usefulness of the paper. We apologise for the delay in getting back, summer travels came in the way. We tried to address most of the concerns, please find the details inline. This included modifying the model part to make it cleared and adding details of the analyses used in reinterpretation. Additionally, we have added a study of a new jet reconstruction technique called the Dynamic Radius Jet Clustering, added plots of a couple of new discriminating variables, and proposed a loose set up cuts for an experiment analysis with a cutflow.
This paper considers a signature of hidden valley models -- models containing a mediator U(1) and a strongly-coupled dark sector containing dark pions -- where dark pions decay predominantly to b-quarks. First, three existing searches are considered which cover this possibility: searches for semi-visible jets; searches for dark matter production in association with b jets; and searches for missing transverse momentum plus several b-jets. The first of these was already available in RIVET, while the second two were implemented by the authors, with validation material presented in this paper.
Secondly, the paper examines the proposed signature in detail and proposes observables which could be useful to devise a full search. This consists of relatively novel techniques such as using variable-radius jets, but also loosening lepton veto requirements. Plots of various kinematic variables for signal and background events are shown.
I believe that the work in the paper is sound and interesting, and may be used to help design a search in future. However, it requires substantial editing, including the abundant typos (even in the figures, e.g. Figure 7 contains Title=... twice).
>>We sincerely apologise for the sloppiness, went over the draft carefully and hopefully fixed them.
In particular:
The introduction is too abrupt and unclear: after one paragraph of motivation, the next starts discussing how signal samples are generated, when it has not been made particularly clear what the signal is or why. I suggest that the signature be spelled out in words clearly, and discuss the various other searches for similar signals at that point before details of simulations.
>>We have restructured this part. The introduction now contains an outline to guide the reader, and we now started the signal generation discussion with the vanilla SVJ first, before moving to our specific signature.
The phrase "which allowed to validate the Rivet routines written for the analyses, if not already been made public." should be clarified to more clearly state at that point which analyses existed and which are novel.
>>The H(bb)+MET and SUSY MultiB routines are not public yet, but we will make them public in the next Rivet release, since the work is done. We modified: if not already been made public ->which are either public or will be soon.
The phrase "Since VR jet radius adaptive compared to RC," needs an "is".
>>Sorry, fixed.
>>"The Yukawa coupling" kappa is not properly defined, nor is it actually used.
We removed the phrase: via the Yukawa coupling, $\kappa$ during the rewrite.
I also have some worries. The first is that the quality of the recast in Table 2 is very poor. The authors seem content that the yields are "mostly within 100%" but this is barely true; such a recast, with only those benchmark points as cross-checks, would not normally be considered validated. They also make the statement that "It is remarkably difficult to reproduce [MET] distributions at particle level" which needs qualification, since I am not aware that this is true. I suppose that the left plot of figure 2 is supposed to show that that analysis is not sensitive to their signal, but I am not sure what the right-hand plot is for, since a casual reader might infer that it is a reproduction of the exclusion curve for validation purposes.
>>Did the referee mean Table 1, which is worse than Table 2? But in general we agree that neither are great, which is a consequence of modeling large MET as we mentioned. For these kinds of signals, large MET comes from two sources: the real MET from the DM candidates, and the angular and energy mismeasurement of jets. While the first can be modeled well in a pure particle level study, the latter cannot be. One can use Delphes to mimic some sort of detector effect, but in our experience it is far from accurate. We used jet energy and angular smearing implemented in Rivet as we mentioned, which somewhat helps, but we think it is fair to point out the shortcoming. We agree that a more detailed validation would have been better, we did run a couple other signal points, which had similar levels of (dis-)agreement. However, we preferred to highlight this point as it was closest to our signal in terms of the yield.
The choice of loosening the lepton veto to 75 GeV is curious, since I understood that leptons along with b-jets were normally removed by isolation requirements prior to any veto, and this is normally regarded as rather efficient. The authors should at least comment on this.
>>Indeed, but it was hard to reproduce isolation in a particle level analysis, so we went with a cut which seemed to work. We added a comment along the lines of your suggestion.
Table 3 presents selection efficiencies for different jet strategies, and are referred to as "It is evident that the use of VR jets will enhance the signal efficiency." But this is only helpful if it enhances signal compared to background.
>> Indeed, we did study the background with VR jets, and no noticeable difference was observed. We added a line in the text.
Finally, I am surprised that the authors shy away from proposing an actual search. Their explanation for this is that they are using particle-level information and detector effects may be important. However, Rivet is supposed to compensate for this, and a public paper would at best be used as a starting point for an experimental study, so the more details that are there the better.
>>The point is well taken, and we added a basic cutflow, with the disclaimer that it will have to be optimised properly in an experimental analysis.

---

## Round 2 · Referee Report · Anonymous (Referee 2) · 2024-6-10

Report
Secondly, the paper examines the proposed signature in detail and proposes observables which could be useful to devise a full search. This consists of relatively novel techniques such as using variable-radius jets, but also loosening lepton veto requirements. Plots of various kinematic variables for signal and background events are shown.
I believe that the work in the paper is sound and interesting, and may be used to help design a search in future. However, it requires substantial editing, including the abundant typos (even in the figures, e.g. Figure 7 contains Title=... twice).
In particular:
The introduction is too abrupt and unclear: after one paragraph of motivation, the next starts discussing how signal samples are generated, when it has not been made particularly clear what the signal is or why. I suggest that the signature be spelled out in words clearly, and discuss the various other searches for similar signals at that point before details of simualations.
The phrase "which allowed to validate the Rivet routines written for the analyses, if not already been made public." should be clarified to more clearly state at that point which analyses existed and which are novel.
The phrase "Since VR jet radius adaptive compared to RC," needs an "is".
"The Yukawa coupling" kappa is not properly defined, nor is it actually used.
I also have some worries. The first is that the quality of the recast in Table 2 is very poor. The authors seem content that the yields are "mostly within 100%" but this is barely true; such a recast, with only those benchmark points as cross-checks, would not normally be considered validated. They also make the statement that "It is remarkably difficult to reproduce [MET] distributions at particle level" which needs qualification, since I am not aware that this is true. I suppose that the left plot of figure 2 is supposed to show that that analysis is not sensitive to their signal, but I am not sure what the right-hand plot is for, since a casual reader might infer that it is a reproduction of the exclusion curve for validation purposes.
The choice of loosening the lepton veto to 75 GeV is curious, since I understood that leptons along with b-jets were normally removed by isolation requirements prior to any veto, and this is normally regarded as rather efficient. The authors should at least comment on this.
Table 3 presents selection efficiencies for different jet strategies, and aare referred to as "It is evident that the use of VR jets will enhance the signal efficiency." But this is only helpful if it enhances signal compared to background.
Finally, I am surprised that the authors shy away from proposing an actual search. Their explanation for this is that they are using particle-level information and detector effects may be important. However, Rivet is supposed to compensate for this, and a public paper would at best be used as a starting point for an experimental study, so the more details that are there the better.
I prepared this report before reading the review of the first referee until this point, and I find that we broadly agree. I would not insist, however, that the authors propose a set of cuts, even if I am surprised that they did not attempt to do so.
Recommendation
Ask for major revision

---

## Round 3 · Referee Report · Anonymous (Referee 1) · 2024-10-8

Report

With their resubmission the authors have substantially improved the quality of the manuscript and addressed several of my main concerns. In particular table 4 constitutes a welcome addition. However, I worry that the most dominant background is missing in the table (and in the figures), namely Z+jets, with the Z boson decaying invisibly. Is there a good argument why this background is not considered? If such an argument can be provided, I would be convinced that the proposed analysis is promising, and that the present work provides sufficient motivation for a more detailed follow-up study to warrant publication in SciPost.

Requested changes

  • Table 4 is labelled "Selection efficiency in %", but the cells contain numbers of events, not their relative change.

Recommendation

Ask for minor revision

  • validity: -
  • significance: -
  • originality: -
  • clarity: -
  • formatting: -
  • grammar: -

Author:  Deepak Kar  on 2024-10-09  [id 4850]

(in reply to Report 1 on 2024-10-08)

We again thank the referee. The referee is of course right, and Z(nunu)bb does have a comparable cross-section compared to ttbar backgrounds, but we saw minphi and high MET requirements do kill it. We think the former is because the Z-boson is balancing the bb system. Please find attached the updated yields table, will the typo about efficiency fixed as well.

Attachment:

---

## Round 3 · Referee Report · Anonymous (Referee 2) · 2024-10-16

Report

The authors have addressed the concerns of my report. They have also added a full-fledged proposal for a new search with background simulations, and this definitely strengthens the paper.

Regarding the quality of the recasts, indeed I must have been referring to table 1. It is a shame that the agreement is still so poor; assuming that there is no mistake in their implementation then the problem will be the perennial issue of efficiencies of the analysis objects, b-tagging, trigger etc. In general sorting this out is what takes most of the time, and where showing a cutflow can be helpful. Of course, that is not an automatic output of RIVET (unlike some other frameworks; similarly other frameworks make the implementation of isolation at particle level straightforward). So I will accept that the authors have done the best job possible given the available materials: readers will be able to use the recast with caution.

Recommendation

Publish (meets expectations and criteria for this Journal)

---

## Round 3 · Author Response

We thank the referees for comprehensive reviews, and making suggestions to improve the quality and usefulness of the paper. We apologise for the delay in getting back, summer travels came in the way. We tried to address most of the concerns.

---

## Round 3 · List of Changes

• Reorganised the model part, switched from NF_1 to NF_2 for theoretical consistency, but none of the conclusions change.
  • Added plots of a couple of new discriminating variables, and proposed a loose set up cuts for an experiment analysis with a cutflow. Also expanded the text to motivate the analysis strategy.
  • We have added a study of a new jet reconstruction technique called the Dynamic Radius Jet Clustering.
  • We added more details on the analyses used for reinterpretation to help the reader.
  • Fixed typos, and textual inconsistencies.

---

## Round 4 · Author Response

We thank the referees for their positive feedback, the requested changes have been done.

---

## Round 4 · List of Changes

The Z(nunu)bb background is added in Table 4, in the Figures 8-10, and in the text, as requested by the referee. Also the Table 4 has been fixed to reflect that the yields are shown.

---

## Editorial Decision

published